# Cost-effectiveness of QuantiFERON-TB Gold In-Tube versus tuberculin skin test for diagnosis and treatment of Latent Tuberculosis Infection in primary health care workers in Brazil

**Rafaela Borge Loureiro**[1,2], **Ethel Leonor Noia Maciel**[2,3], **Rosangela Caetano**[4], **Renata Lyrio Peres**[2,5], **Geisa Fregona**[2], **Jonathan E. Golub**[6], **José Ueleres Braga**[1]*

**1** Department of Epidemiology, Institute of Social Medicine (IMS), Universidade do Estado do Rio de Janeiro (UERJ), Rio de Janeiro, RJ, Brazil, **2** Laboratory of Epidemiology (Lab-Epi), Universidade Federal do Espírito Santo (UFES), Vitória, ES, Brazil, **3** Graduate Program in Collective Health (PPGSC), Universidade Federal do Espírito Santo (UFES), Vitória, ES, Brazil, **4** Department of Health Policy, Planning and Administration, Institute of Social Medicine (IMS), Universidade do Estado do Rio de Janeiro (UERJ), Rio de Janeiro, RJ, Brazil, **5** Center of Infectious Diseases (NDI), Universidade Federal do Espírito Santo (UFES), Vitória, ES, Brazil, **6** Division of Infectious Diseases, Bloomberg School of Public Health, Johns Hopkins University, Baltimore, Maryland, United States of America

* ueleres@gmail.com

## Abstract

### Objectives

The goal of this study was to perform a cost-effectiveness analysis from the public health system perspective, comparing five strategies for Latent Tuberculosis Infection (LTBI) diagnosis in primary health care workers in Brazil.

### Design

Analytical model for decision making, characterized by cost-effectiveness analysis.

### Setting

Primary Care Level, considering primary health care workers in Brazil.

### Participants

An analytical model for decision making, characterized by a tree of probabilities of events, was developed considering a hypothetical cohort of 10,000 primary health care workers, using the software TreeAge Pro™ 2013 to simulate the clinical and economic impacts of new diagnostic technology (QuantiFERON®-TB Gold in-Tube) versus the traditional tuberculin skin test.

**Data Availability Statement:** All relevant data are within the manuscript and its Supporting Information files.

**Funding:** This study was supported by CNPq (Conselho Nacional de Desenvolvimento Científico e Tecnológico) MCT/CNPq No. 14/2009 – Universal and by the International Clinical, Operational and Health Services Research Training Award grant – ICOHRTA [AIDS/TB]. None of the funding agencies is responsible for the statements in this article. The funders had no role in study design, data analysis, decision to publish, or preparation of the manuscript.

**Competing interests:** The authors have declared that no competing interests exist. This study has not been submitted elsewhere for publication. All authors have approved the manuscript.

## Methods

This model simulated five diagnostic strategies for LTBI in primary health care workers (HCW) in Brazil: tuberculin skin testing using ≥5 mm cut-off, tuberculin skin testing ≥10 mm cut-off, QuantiFERON®-TB Gold in-Tube, tuberculin skin testing using ≥5 mm cut-off confirmed by QuantiFERON®-TB Gold In-Tube if TST positive, tuberculin skin testing using ≥10 mm cut-off confirmed by QuantiFERON®-TB Gold In-Tube if TST positive.

### Primary and secondary outcome measures

The outcome measures are the number of individuals correctly classified by the test and the number of Tuberculosis cases avoided.

### Results

The most cost-effective strategy was the tuberculin skin test considering ≥10mm cut-off. The isolated use of the QuantiFERON®-TB Gold In-Tube revealed the strategy of lower efficiency with incremental cost-effectiveness ratio (ICER) of US$ 146.05 for each HCW correctly classified by the test.

### Conclusions

The tuberculin skin test using ≥10 mm cut-off was the most cost-effective strategy in the diagnosis of Latent Tuberculosis Infection in primary health care works in Brazil.

## Introduction

Health Care Workers (HCWs) are one of the most vulnerable groups to infection by *Mycobacterium tuberculosis* (Mtb) [1]. The resurgence of tuberculosis (TB) in the world, between the 1980s and 1990s, was accompanied by many nosocomial outbreaks, with several deaths of these professionals[2, 3]. Thus, TB screening tests in these workers are considered essential for identification of latent and active disease in an effort to reduce transmission in health services[4].

According to estimates from the World Health Organization (WHO), 10.0 million people (range, 9.0–11.1 million) developed TB disease in 2017 and 1.6 million death due to TB occurred[5].Despite the decline in the incidence and mortality rates of the disease, one-third of the world population has Latent Tuberculosis Infection (LTBI)[6]. This situation may have improved, as up-to-date estimates indicate that about a quarter of the world's population is infected, corresponding to 1.7 billion people[7]. To achieve the United Nations Sustainable Development Goal of eliminating this endemic disease by 2050 is necessary to diagnose and to treat the disease with new approaches.

A strategy indicated to increase control is by detection and treatment of LTBI[8] because infected persons have a 10% risk to develop active TB during life[9]; this risk can be reduced with the use of isoniazid (INH) preventive therapy.

In Brazil, the Ministry of Health (MOH) reported 72,788 new cases of TB in 2018, with an incidence rate of 34.8/100,000 inhabitants. Although an average annual decrease of 1.0% was observed between 2009 and 2018, the incidence coefficient increased in 2017 and 2018 compared to 2014–2016, indicating the need to improve control measures in Brazil[10]. TB mortality is also declining. In 2001, the mortality rate for the country was 3.1 deaths/100,000 inhabitants, falling to 2.2/100,000 inhabitants in 2016 and 2017 [10]. Despite the decrease in

the number of cases, Brazil is ranked among the 30 high TB burden countries by the WHO, which account for 80% of TB worldwide[11].

The identification of people with LTBI is considered by WHO as a priority in controlling disease[8], especially in developing countries where the incidence of the active disease has shown a reduction. In Brazil, the National Tuberculosis Control Program (NTCP) includes health professionals in the category of highest risk[12] due to their occupational exposure[4]. Similarly to the Brazilian medical associations[13] and the WHO[14], the NTCP recommended measures to reduce the risk of transmission of the infection in centers for tuberculosis diagnosis and treatment in the country.

The tuberculin skin test (TST), standard method commonly used for the diagnosis of LTBI, has low cost. However, it is limited by low specificity due to false-positive results in populations vaccinated with Bacillus Calmette-Guérin (BCG) or infected with Nontuberculous Mycobacteria (NTM), not overcome by using higher cut-offs as test positivity criteria. Another difficulty of this test is the need for repeated visits to the health system for their achievement and reading, influencing adherence to screening[15–18].

In order to address the challenges posed by the TST, a new diagnostic technology has been introduced as tests for LTBI, the interferon-gamma release assays (IGRA). These require only one visit to the clinic, with results available within a short period of time (24 hours) and are not subject to the subjectivity of reading. One of these tests based on antigen detection in whole blood is QuantiFERON-TB Gold In-Tube (QTF-GIT), an in vitro immunoassay using an ensemble of peptides simulating the ESAT-6, CFP-10 and TB7.7 (p4) proteins[16–19]. This assay, the only IGRA that is already approved by the Brazilian Health Surveillance Agency (ANVISA) for marketing and use in Brazil, has high sensitivity and specificity and differentiates LTBI of immune response to vaccination and infection by NTM[20–22]. Despite the requirement for laboratory infrastructure, equipment and supplies are expensive[15, 22, 23], its use for this indication was evaluated as cost-effective in various locations and incorporated into TB control guidelines in some developed countries[24, 25].

In Brazil, a study published in 2013[26] assessed the cost-effectiveness in the public health system perspective, comparing three strategies for the diagnosis of LTBI in immunocompetent adults who were close contacts of TB cases: TST, QFT-GIT, and QFT-GIT in individuals with a positive TST. Contrary to the results of studies conducted in other countries, the TST strategy was more cost-effective, with an incremental cost-effectiveness ratio (ICER) of US$ 16 per case of TB prevented. Gaps persist in the literature, especially with regard to its use in specific populations, such as HCWs. There is also a lack of economy studies for evaluating the effectiveness of these tests in health professionals of countries with high load and wide coverage for BCG, such as Brazil.

Thus, conducting economic analysis of diagnostic methods for LTBI in primary HCWs is timely and relevant and can support decision-making processes related to the incorporation of the new test in the country. The aim of this study was to evaluate the cost-effectiveness of TST versus QFT-GIT in the diagnosis and treatment of LTBI in primary HCWs, in the perspective of the Brazilian Unified Health System (*Sistema Único de Saúde*–SUS), comparing five strategies that include the QFT-GIT, distinct cut-off points for TST and sequential use of the two tests.

## Methods

### Model structure

A cost-effectiveness evaluation was conducted considering a hypothetical cohort of 10,000 HCWs of both sexes working in primary care.

An analytical model for decision making, characterized by a tree of probabilities of events, was developed using the Tree-Age ProTM 2013 (TreeAge Software Inc, Williamstown, MA, USA) to simulate the natural history of LTBI, clinical outcomes and economic impacts of the new diagnostic technology (QFT-GIT) versus the traditional TST. This model simulated five diagnostic strategies for detection and treatment of LTBI: (1) TST using a ≥5 mm cut-off point, which is recommended by the National Tuberculosis Control Program in asymptomatic adults contacts; (2) TST using a ≥10 mm cut-off point, which is currently recommended by the NTCP for the management of latent infection in health care workers; (3) QFT-GIT test; (4) TST with a ≥5 mm cut-off point, followed by QFT-GIT when TST test positive; and (5) TST with a cut-off point of ≥10 mm, followed by QFT-GIT when the TST was positive, a strategy that has proven to be more cost-effective in some countries of TB high burden[25, 27] and that has been recommended by some guidelines[24, 28, 29]. Fig 1 and Fig 2 show the analytical decision model developed.

**Patient and public involvement.** A cost-effectiveness analysis was conducted considering a hypothetical cohort of 10,000 HCWs of both sexes working in primary care; patients and/or public were not involved.

**Measures of effectiveness.** Two measures of effectiveness were chosen. The first refers to the number of individuals correctly classified as with infection and was calculated by the sum of truepositives and true negatives, considering the prevalence of LTBI estimated for HCWs. The other measure is the number of TB cases avoided from diagnosis of LTBI and its preventive treatment with INH.

The time horizon of the study was restricted to one year.

**Model assumptions.** The strategies simulated in the analytical model are based on the screening process and treatment of the TB recommended by the WHO[14], the Brazilian Society of Pulmonology and Phthisiology[13], and the NTCP[30].

The trajectory of health primary care professionals was modeled by submitting each screening strategy study and clinical consequences of treatment decisions that resulted from this. For each of the strategies tested, probabilities of infection and detection LTBI were estimated considering whether the infection was recent or remote (≥2 years). In professionals where infection had been detected, the risk of active TB was modeled with and without treatment of the infection. Adherence to treatment and risks of preventive therapy were also considered with INH for 6 months as recommended by NTCP.

As a reference case, complete adherence to treatment was considered and associated risks were limited to severe hepatotoxicity; there was still the possibility of death by Drug-Induced Liver Injury (DILI). Finally, mortality and treatment effectiveness in patients with active TB were considered.

Whereas the simulation model of the natural history of LTBI consists of a simplification of reality, some assumptions have been made, taking as a reference the coherence with clinical and epidemiological knowledge available. Thus, this study assumed that: (a) all HCWs were asymptomatic, without active TB; (b) there were no HIV-infected individuals; (c) 20% of HCWs with LTBI and positive TST had recent infection; (d) all professionals diagnosed with LTBI received prophylactic treatment with isoniazid prescription after clinical examination and X-rays of the chest, excluding diagnosis of active TB; (e) all cases of active TB and LTBI would be sensitive to antituberculosis drugs; and (f) HCWs who developed DILI would complete only three months of treatment and this time therapy was associated with an increased risk of progression of LTBI to active disease, compared to those with complete treatment.

## Model parameters

Clinical, epidemiological and test accuracy parameters derived from original articles, systematic reviews and studies conducted in Brazil on the technologies; they are shown in Table 1.

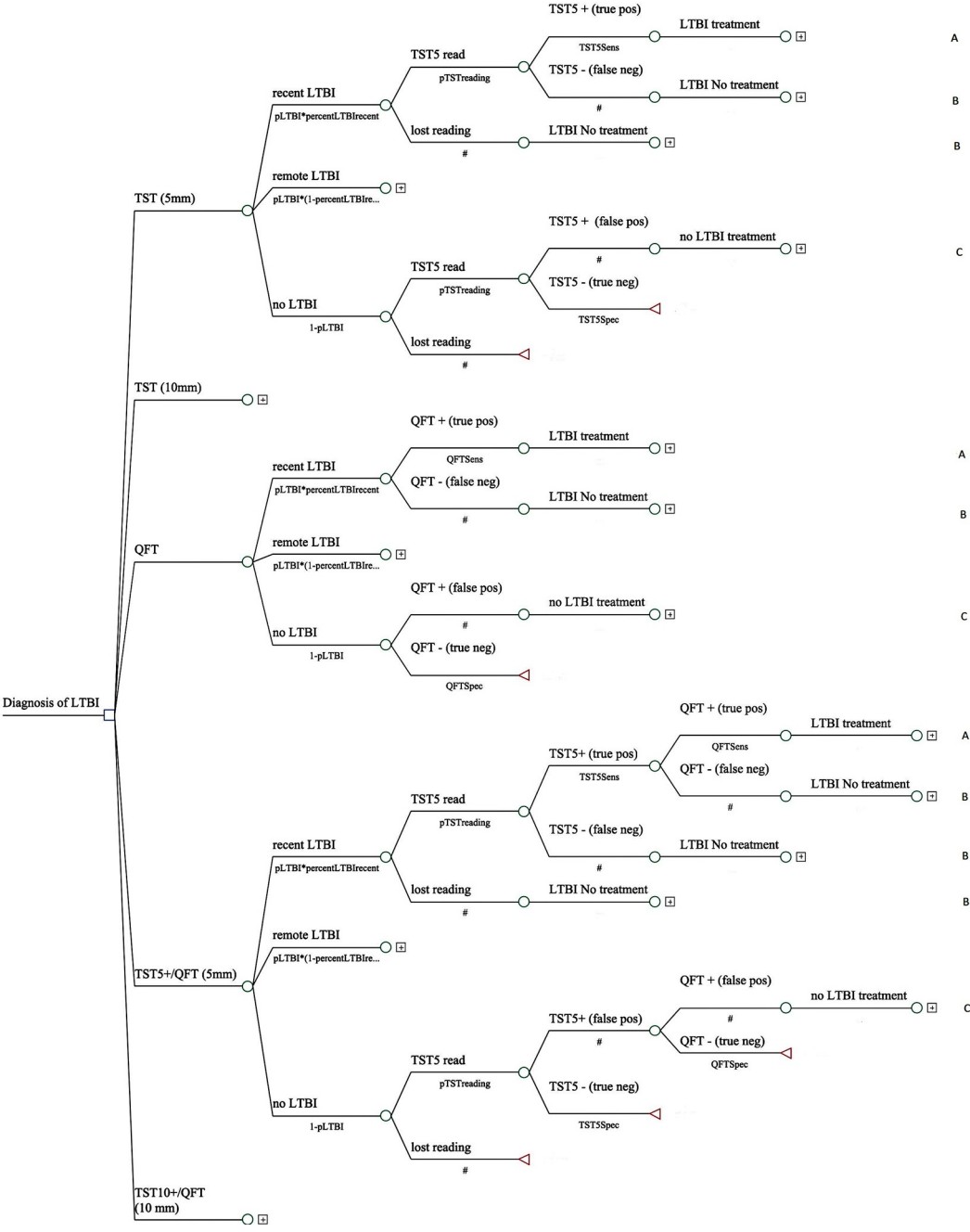

**Fig 1. Decision tree for diagnosis of Latent TB Infection (LTBI).**

**Clinical and epidemiological parameters.** The estimated prevalence of LTBI was obtained from the partial results of the Brazilian study "INATA: Infecção e Adoecimento por Tuberculose entre Profissionais de Saúde da Atenção Básica" (INATA: Infection and Illness by Tuberculosis among Primary Health Care Workers). This survey, conducted from January 2011 to December 2013, assessed the prevalence of LTBI in primary care professionals. The sample included 708 health professionals, with 20–70 years, selected from cities with high Tb incidence from the five geographical regions: Manaus (93.3/100,000), Salvador (62.3/100,000),

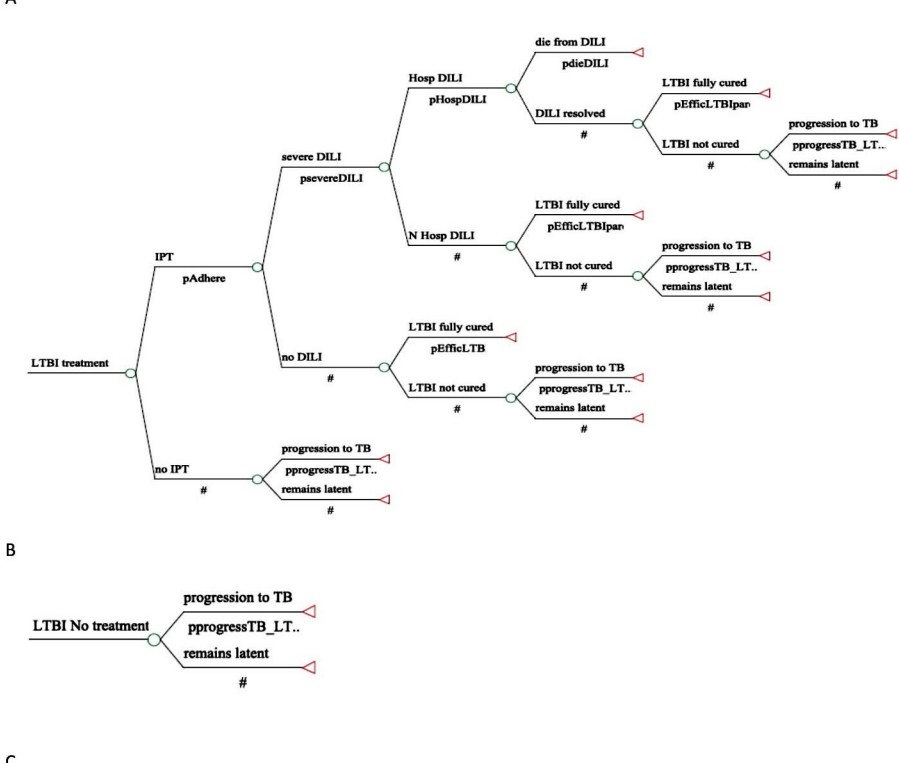

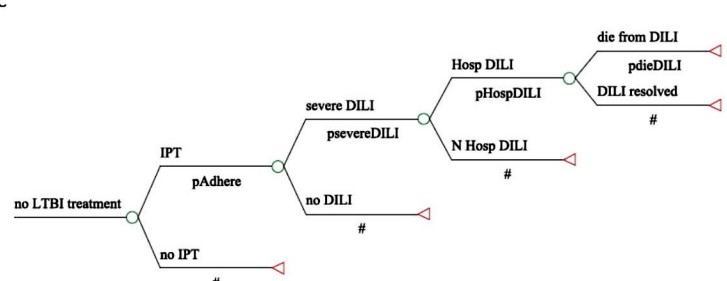

**Fig 2. Decision tree for diagnosis of Latent TB Infection (LTBI) (continued from Fig 1). Notes:** (A) LTBI treatment, (B) LTBI No treatment, (C) No LTBI treatment.

Cuiabá (116.5/100,000), Vitória (46.2/100,000) and Porto Alegre (99.5/100,000) [10], submitted to diagnostic tests TST and QuantiFERON-TB Gold In-Tube. The overall prevalence estimated of LTBI, based on TST ≥10 mm cut-off point, was 0.40 (95% CI 0.36–0.44).

The proportion of professionals who returned to the TST reading used in the reference case was 0.96 (INATA Project finding) and as limits of the variation range (0.95–0.98) corresponding the highest and lowest proportion of returns from the municipalities of the research (data based on personal correspondence with project INATA coordinator).

Adherence rates to treatment used in the reference case based on the results of the Brazilian site of a multicenter randomized controlled clinical study, which investigated the predictors of adherence to treatment of LTBI in adults[31]; range of values used in the sensitivity analysis were obtained from the literature. Estimates of treatment adherence rate in health workers diagnosed as LTBI by QTF were not available in the literature at the time of the study, and the same parameter and range above were used.

**Table 1. Clinical and epidemiological parameters, accuracy of tests and cost parameters used in the model.**

| Clinical and Epidemiological Parameters | Base-Case Probability | Range[a] | Source |
|---|---|---|---|
| Prevalence of LTBI | 0.40 | 0.36–0.44 | INATA |
| Probability of recent LTBI | 0.20 | 0.05–0.50 | Estimated |
| Probability of return to TST reading | 0.96 | 0.95–0.98 | INATA |
| Adherence to LTBI treatment (6 months)[b] | 0.53 | 0.47–0.78 | Jasmer, 2002[41]; LoBue, 2003[42]; Trajman et al., 2010[31]; Horsburgh Jr, 2010[43]; IUAT, 1982[32] |
| Efficacy of LTBI treatment (6 months) | 0.65 | 0.50–0.93 | IUAT, 1982[32]; Khan, 2002[44]; Comstock, 1999[45] |
| Efficacy of LTBI treatment (3 months) | 0.21 | 0.10–0.31 | IUAT, 1982[32]; Khan, 2002[44]; Comstock, 1999[45] |
| Probability of DILI related to LTBI treatment | 0.001 | 0.00001–0.01 | Nolan et al., 1999[46]; Steele, 1991[47]; Fountain, 2005[48]; Saukkonen et al., 2006[49]; Linas et al., 2011[50] |
| Probability of hospitalization due to DILI | 0.00015 | 0.00010–0.00020 | Saukkonen et al, 2006[49]; Leung et al., 2011[51]; Nolan et al., 1999[46] |
| Probability of deaths by DILI | 0.00001 | 0.000001–0.0003 | Millard, 1996[52]; Salpeter et al., 1997[53]; Saukkonen et al., 2006[49]; Leung et al., 2011[51] |
| Evolution of recent LTBI to TB, with complete treatment for LTBI | 0.005 | 0.0047–0.015 | IUAT Trial, 1982[32] |
| Evolution of recent LTBI to TB, with partial treatment for LTBI | 0.0113 | 0.0094–0.015 | IUAT Trial, 1982[32] |
| Evolution of recent LTBI to TB, without treatment for LTBI | 0.08 | 0.05–0.10 | Pai, 2008[35]; Vynnycky E, Fine PE, 1997[34]; Public Health Agency of Canada, 2014[54] |
| Evolution of remote LTBI to TB, without treatment for LTBI | 0.04 | 0.025–0.05 | D'Arcy, 1972[35]; Vynnycky E, Fine PE, 1997[34]; Public Health Agency of Canada, 2014[54] |
| **Accuracy of Test Parameters** | **Base-Case Probability** | **Range[a]** | **Source** |
| TST sensitivity (.≥ 5 mm) | 0.82 | 0.68–0.97 | Diel et al., 2007[55] |
| TST specificity (.≥ 5 mm) | 0.61 | 0.35–0.79 | Diel et al., 2007[55]; Lee et al. 2006[56] |
| TST sensitivity (.≥ 10 mm) | 0.77 | 0.55–0.95 | Menzies, 2007[17]; Pai, 2008[35] |
| TST specificity (.≥ 10 mm) | 0.59 | 0.43–0.73 | Pai, 2008[35]; Diel, 2011[37] |
| QFT-GIT sensitivity | 0.78 | 0.54–0.82 | Pai, 2008[35]; Menzies, 2007[17]; Diel, 2010[39]; Zwerling, 2012[36] |
| QFT-GIT specificity | 0.98 | 0.96–0.99 | Menzies, 2007[17]; Pai, 2008[35]; Diel, 2009[38]; Diel, 2010[39]; Diel 2011[37] Zwerling, 2012[36] |
| Probability of indeterminate QFT-GIT result | 0.05 | 0.02–0.09 | Shahidi, 2012[57]; Metcalfe, 2011[58] |
| **Cost Parameters** | **Base-Case Value (US$)** | **Range[a]** | **Source** |
| *Costs Related to LTBI Diagnosis* | *12.55* | *8.29–16.81* | |
| Initial medical consultation | 8.51 | 4.25–12.76 | MOH/SIGTAP |
| Chest radiograph | 4.04 | - | MOH/SIGTAP |
| *Costs Related to Active TB Diagnosis* | *16.12* | *6.04–18.12* | |
| Initial medical consultation | 8.51 | 4.25–12.76 | MOH/SIGTAP |
| Chest radiograph | 4.04 | - | MOH/SIGTAP |
| Sputum smear | 3.57 | 1.79–5.36 | MOH/SIGTAP |
| *Cost of QFT-GIT:* | *39.00* | *31.77–47.66* | *Estimated* |
| Human Resources [c] | 2.24 | 1.79–2.68 | |
| QFT-GIT test kit | 33.88 | 27.10–40.65 | |
| Consumables[d] | 1.81 | 1.81–2.71 | |
| Equipment[e] | 1.07 | 1.07–1.60 | |
| *Cost of TST:* | *7.62* | *6.96–9.54* | *Estimated* |
| Human resources[f] | 2.12 | 1.48–2.54 | |
| PPD RT23 2 UT/1.5 ml | 4.14 | 4.14–4.97 | |
| Consumables[g] | 1.31 | 1.31–1.97 | |

*(Continued)*

**Table 1.** (Continued)

| | | | |
|---|---|---|---|
| Equipment[h] | 0.05 | 0.02–0.05 | |
| *Cost of Complete LTBI Treatment (6 months)* | *42.66* | *30.69–59.76* | |
| Isoniazid (300 mg/day) | 5.95 | - | **MOH/NTBP** |
| Blood count | 3.50 | 1.75–5.24 | MOH/SIGTAP |
| Serum dosage GOT | 1.71 | 0.85–5.13 | MOH/SIGTAP |
| Serum dosage GPT | 1.71 | 0.85–5.13 | MOH/SIGTAP |
| Medical consultation | 29.79 | 21.28–38.30 | MOH/SIGTAP |
| *Cost of Partial LTBI Treatment (3 months)* | *35.90* | *28.19–43.62* | |
| Isoniazid (300 mg/day) | 4.25 | - | **MOH/NTBP** |
| Blood count | 5.25 | 3.50–6.99 | MOH/SIGTAP |
| Serum dosage GOT | 2.56 | 1.71–3.42 | MOH/SIGTAP |
| Serum dosage GPT | 2.56 | 1.71–3.42 | MOH/SIGTAP |
| Medical consultation | 21.28 | 17.02–25.53 | MOH/SIGTAP |
| *Cost of DILI* | *254.63* | *184.90–294.05* | |
| Costs of hospitalization | 239.20 | 177.19–265.78 | MOH/SIGTAP |
| Medical consultation | 8.51 | 4.25–12.76 | MOH/SIGTAP |
| Blood count | 3.50 | 1.75–5,25 | MOH/SIGTAP |
| Serum dosage GOT | 1.71 | 0.85–5.13 | MOH/SIGTAP |
| Serum dosage GPT | 1.71 | 0.85–5.13 | MOH/SIGTAP |
| *Cost of Death by DILI[i]* | *432.88* | *216.44–650.59* | |
| Costs of hospitalization in ICU III[i] | 432.88 | 216.44–650.59 | MOH/SIGTAP |
| *Cost of Active TB Treatment (6 months)* | *77.27* | *49.65–102.21* | |
| RHZE scheme (2 months) | 13.44 | - | **MOH/NTBP** |
| RHZ scheme (4 months) | 8.18 | - | **MOH/NTBP** |
| Medical consultation | 29.79 | 12.76–34.04 | MOH/SIGTAP |
| Sputum smear | 12.51 | 5.36–12.51 | MOH/SIGTAP |
| Chest radiograph | 4.04 | 4.04–8.08 | MOH/SIGTAP |
| Culture for sputum smear | 2.39 | 2.39–10.43 | MOH/SIGTAP |
| Blood count | 3.50 | 1.75–5.25 | MOH/SIGTAP |
| Serum dosage GOT | 1.71 | 0.85–5.13 | MOH/SIGTAP |
| Serum dosage GPT | 1.71 | 0.85–5.13 | MOH/SIGTAP |

Caption: DILI–Drug Induced Liver Disease, LTBI–Latent Tuberculosis Infection; PPD–purified protein derivative; RHZE—Rifampicin, Isoniazid, Pyrazinamide and Ethambutol; RHZ—Rifampicin, Isoniazid, Pyrazinamide; TST–Tuberculosis Skin Test; QFT-GIT–QuantiFERON-TB Gold In-Tube; SIGTAP–Procedure Table Management System, medications and OPM of SUS; SIASG–General Services Management System; TB–Tuberculosis; GOT–glutamic oxaloacetic transaminase/aspartate aminotransferase; GPT–glutamate pyruvate transaminase/alanine aminotransferase; NTBP–National Tuberculosis Program; MOH–Brazilian Ministry of Health.

Notes:

[a]–Assumes that individuals had adherence to LTBI treatment with INH 300 mg/day.

[b]–The parameter variation range was set based on the upper and lower limits in relation to the reference case.

[c]–Nursing staff time, laboratory technician time

[d]–Gloves, needles, tourniquet, cotton, alcohol, box for syringes, eppendorf, cryotube, color-coded insert (red and blue), DNA Free Pyrogen (200 μl–sterile and with filter), D1000 Diamond®Tipack 100–1000 μl (sterile and with filter)

[e]–incubator, centrifuge, microplate washer, microplate reader, computer, printer

[f]–Nursing staff time

[g]–Gloves, cotton, alcohol, syringes with needles, box for syringes, thermic box and ice bag

[h]–Fridge, thermometer with alarm, millimeter ruler

[i]–It was considered that the cost of ICU type III was for severe hepatotoxicity. US$ 1.00 = R$ 2.35 (mean exchange rate in 2014).

LTBI treatment efficacy was based on the trial conducted by the International Union Against Tuberculosis (The Union) to evaluate the effect of prophylactic regimens with isoniazid[32]. The results of the schemes were used with duration of three and six months, with the values applied in reference case corresponding for all participants. The Union trial[32] also estimated the rates of progress to active TB after complete (6 months) and partial (3 months) treatment of LTBI.

The probability of progression of LTBI to active TB in individuals without treatment varies according to their age and post-exposure time to MTb. Classic studies of tuberculosis on progression findings were used[33], including those that evaluated the age effect on disease risk [34].

The probability of occurrence of DILI as well as hospitalization and death due to this condition also had international literature as source. DILI parameters took into account serious hepatotoxicity only.

**Test accuracy parameters.**   The proportion of undetermined results for strategies that included sequential or QFT-GIT alone was obtained from the literature, taking into account adult and HIV-uninfected subjects.

The parameters of test accuracy were drawn from systematic reviews and meta-analysis studies published by Menzies[17], Pai[35], Zwerling[36] and Diel[37–39].

**Cost parameters.**   The costs of screening and treatment of LTBI were analyzed from the perspective of the Brazilian Health System as responsible for financing the diagnosis and treatment of tuberculosis in Brazil. All costs were converted to U.S. dollars (US$) at the rate of 2.35 reais/1 US$, the average conversion rate for 2014. No discount was applied because of the short horizon of the study (one year). There was no inflation rate adjustment.

Direct medical costs related to the detection of cases of LTBI examined by diagnostic strategies were assessed, as well as those arising from assistance to detected cases and losses within each approach.

Until the preparation of this paper, the QFT-GIT was not incorporated into the SUS payment procedures tables. Moreover, currently, TST costs are covered in the basic care actions for TB control. Thus, an estimate of costs of these two exams was made to fill the model used. For cost estimates, stages of the production process of the two procedures were identified based on the observation of 164 TST tests and 640 QFT-GIT assays performed from January to March 2013, in the Laboratory of Immunology (LI) of the Center Infectious Diseases at the Universidade Federal do Espírito Santo. This study quantified all supplies used in the production of the tests and then assigned monetary values to them based on effective consumption or on estimate of the cost of items under service provider's perspective[40].

This study also considered the costs of supplies (consumption of reagents and materials, as examination gloves, needles syringes, tourniquet, cotton, alcohol, box for syringes), and of equipment (fridge for purified protein derivative–PPD–storage, thermometer with alarm, millimeter ruler for reading PPD, incubator, centrifuge, microplate washer, microplate reader, computer, printer) and the necessary human resources (nurses and laboratory technicians). Financial and accounting data for valuation of supplies and resources used were obtained from the LI and Health Department of the Municipality of Vitória/Espírito Santo.

In addition to the cost of the tests, the cost of screening for LTBI included two medical appointments and a chest X-ray when the TST or QFT-GIT was positive, for active TB exclusion. The number of appointments involved in this exclusion of diagnosis varied in the sensitivity analysis (SA) from one to three appointments to the lower and upper limit, respectively, considering the tests performed.

Isoniazid used to treat LTBI and the quadruple regimen used to treat active TB are centrally purchased by the Brazilian Ministry of Health and dispensed free of charge to patients

according to NTBP recommendations. The complete treatment costs for LTBI included INH regimen with 300 mg/day for six months, and seven medical monitoring visits and three additional tests of hepatotoxicity control: blood count; glutamic oxaloacetic transaminase/aspartate aminotransferase (GOT); and glutamic pyruvic transaminase/alanine aminotransferase (GPT). In the sensitivity analysis, the number of follow-up medical consultations varied around two for less or for more from the values of the reference case, and the number of complementary tests ranged from one to three tests for the lower and upper limit, respectively, considering the clinical evolution of the individual.

The partial treatment costs for LTBI were considered for those who developed hepatotoxicity and did not complete the prophylactic therapy. The regimen included costs with INH 300 mg/day for three months, five medical consultations and three additional tests as mentioned above. Medical monitoring visits and tests varied in sensitivity analysis for one less and one more of the values of the reference case.

Also, cases of severe DILI incurred in hospitalization costs, valued by the code "Treatment of Liver Diseases" present in the Brazilian Hospital Information System table, with a 35% increase in values related to professional care in hospitals considered as Type II Urgency.

Also, cases with severe DILI incurred in hospitalization costs, valued by the code "Treatment of Liver Diseases" present in the Brazilian Hospital Information System (SIH-SUS) table, with a 35% increase in values related to professional and service charges in hospitals considered as Hospital Type II Urgency. In the sensitivity analysis, these values varied to the following extremes: 50% increase, equivalent to hospitalization in Hospital Type III Urgency, for the upper limit; and base value of the hospitalization to the lower limit.

In cases of DILI that evolved to death, costs equivalent to two daily hospitalizations to the intensive care unit adult (ICU III) were included, ranging from 1 to 3 in sensitivity analysis. For those who survived, monitoring costs of severe DILI were also included, resulting in the increase of doctors' visits (varying from one to three consultations in the SA), and two blood count testings, GOT and GPT (ranging from one to three tests in the SA).

For active TB treatment costs that could arise from the screening for LTBI, the basic quadruple regimen recommended by the NTBP was used for six months (rifampicin, isoniazid, pyrazinamide and ethambutol daily for two months, followed by rifampicin, isoniazid and pyrazinamide daily for 4 months). The costs added were related to seven doctor's appointments (varying from three to eight in the SA), five smears (ranging from three to seven), three chest x-ray examinations (ranging from one to seven), complete blood count, two GOT and GPT tests (SA: 1–3), and one sputum culture.

In line with the public health system perspective adopted, the costs of medical appointments, monitoring tests, and hospitalizations were valued according to the payment tables of procedures present in the Brazilian Health System (SIGTAP–*Sistema de Gerenciamento da Tabela de Procedimentos, Medicamentos e OPM do SUS*), considering the tables valid in March 2013. The cost parameters are shown in Table 1.

**Incremental cost effectiveness ratio (ICER).** The comparative efficiency of the diagnostic alternatives for LTBI was measured by the incremental cost-effectiveness ratio, defined as the ratio between the additional cost of the strategy and the clinical effectiveness (estimated for each of the two measures of interest defined above) compared to alternative lower cost-effectiveness ratio. This ratio corresponds to the division of the cost for the observed effect, as set forth below.

$$CER = \frac{Differences\ in\ the\ costs\ between\ the\ alternatives\ A1\ and\ A2}{Differences\ in\ the\ effectiveness\ between\ alternatives\ A1\ and\ A2}$$

For each of the outcome measures two analytical approaches were performed. In one, the TST with ≥10 mm cut-off point was chosen as the base strategy because this cut-off is currently recommended by NTBP in management and decision-making concerning health professionals. In the other, due to possible changes in cut-off points of TST for LTBI diagnosis in health workers, ICER of all strategies is also calculated with each other, from the hierarchy of strategies according to their costs. Dominated strategies (less effective and more costly) and those with weak or extended dominance were eliminated.

No discount was applied due to the short horizon of the study, as recommended in the Brazilian Methodological Guidelines for Economic Evaluation Studies of Health Technologies [59].

**Sensitivity analysis.** Usually, mathematical models incorporate some degree of uncertainty related to the established assumptions and the necessity of assigning values to different parameters necessary to simulate the evolution of the modeled cohort. Sensitivity analysis was used to evaluate the model results considering alternative scenarios to the reference case. To this end, deterministic univariate analysis was performed.

Tornado Diagram was developed to assess the impact of the measures, by varying the following parameters: prevalence of LTBI; risk of progression of recent and remote LTBI to active TB; sensitivity and specificity of TST and QFT-GIT; adherence to treatment of LTBI; efficacy of treatment with INH for LTBI; LTBI and TB treatment costs; costs of TST and QFT-GIT. The variations in the range used are shown in Table 1.

This economic analysis held for aggregate secondary data available in public databases or information coming from literature required no prior approval by the Institutional Review Board (IRB)[30]. The data for the INATA study, approved by the IRB of UFES under number 007/10 were obtained with the authorization of Coordinator of Project–Prof. Ethel Leonor Noia Maciel.

## Results

### Base case

**Number of individuals correctly classified by the tests.** TST represented the lowest cost strategy, whereas the single use of the new QFT-GIT technology for LTBI diagnosis in the population of HCWs was that accounted for higher costs for the public health system. In contrast, the number of individuals correctly classified by the QFT-GIT was the highest, substantially surpassing the other.

The most cost-effective strategy corresponded to the use of TST with the cut-off point recommended currently (≥10 mm) by NTBP for the management of suspect of LTBI among HCWs, at a cost of US$ 4.70 per individual classified correctly (Table 2). The minimal additional cost represented by the use of TST with ≥5 mm cut-off point or more (US$ 1.25) requires further exploration in the sensitivity analysis. Incorporating the unique use of QFT-GIT use alone would result in an additional cost of more than US$ 50.07 compared to the diagnostic strategy currently recommended by the Brazilian Program (Table 2).

The sequential use of the QFT after TST with ≥10 mm cut-off point is dominated (higher cost and lower effectiveness). The strategy with single use of TST≥10 mm continued representing the most cost-effective strategy. Again, minimal differences were found in additional cost by individuals classified with TST ≥5 mm. The single use of the QFT-GIT was the strategy of lower efficiency, with an additional cost by professional correctly classified of over US$ 146.05 (Table 2).

**Number of cases of active tuberculosis avoided.** TST with ≥10 mm cut-off point was the most cost-effective strategy for LTBI diagnosis. The TST strategy using the ≥5mm cut-off has

**Table 2. Cost-effectiveness of screening strategies for diagnosis of Latent Tuberculosis Infection using the number of individuals correctly classified by the tests as measure of effectiveness.**

| Strategies | Total Cost (US $) | Incremental Cost (US $) | Effectiveness | Incremental Effectiveness | Cost/ individuals correctly classified (US$) | ICER (US$) |
|---|---|---|---|---|---|---|
| Comparison of diagnostic strategies for LTBI in relation to the strategy TST≥10mm[a] | | | | | | |
| TST≥10 mm | 30,084.47 | _ | 6,395 | _ | 4.70 | _ |
| TST≥5 mm | 30,471.02 | 386.55 | 6,704 | 309 | 4.54 | 1.25 |
| TST≥5 mm/ QFT-GIT | 46,897.44 | 16,812.97 | 8,222 | 1,827 | 5.70 | 9.20 |
| *TST≥10 mm/ QFT-GIT* | *47,376.73* | *17,292.26* | *8,069* | *1,674* | *5.87* | *dominated* |
| QFT-GIT | 160,531.88 | 130,447.41 | 9 | 2,605 | 17.83 | 50.07 |
| Comparison of diagnostic strategies for LTBI in relation to each other[b] | | | | | | |
| TST≥10 mm | 30,084.47 | _ | 6,395 | _ | 4.70 | _ |
| TST≥5 mm | 30,471.02 | 386.55 | 6,704 | 309 | 4.54 | 1.25 |
| TST≥5 mm/ QFT-GIT | 46,897.44 | 16,426.42 | 8,222 | 1,518 | 5.70 | 10.82 |
| *TST≥10 mm/ QFT-GIT* | *47,376.73* | *479.29* | *8,069* | *-153* | *5.87* | *dominated* |
| QFT-GIT | 160,531.88 | 113,634.44 | 9 | 778 | 17.83 | 146.05 |

Caption: ICER–Incremental Cost-Effectiveness Ratio; LTBI–Latent Tuberculosis Infection; QFT-GIT–QuantiFERON-TB Gold In-Tube; TST–Tuberculosis Skin Test.

Notes:

a ICER is estimated in relation to TST strategy with a ≥10 mm cut-off point, only the dominated strategy was demarcated without removing it from the table.

b ICER is estimated by comparison of the diagnosis strategies for LTBI with each other; Incremental Costs were calculated by the difference between the cost of the strategy and the cost of its previous strategy. Incremental Effectiveness was calculated by the difference between the effectiveness of the strategy and the effectiveness of its previous strategy. US$ 1.00 = R$ 2.35 (mean exchange rate in 2014).

slightly higher effectiveness compared to the ≥10 mm cut-off, at the cost of over US$ 77.00 per additional case avoided. Strategies using QFT-GIT were dominated due to its higher cost and lower effectiveness (Table 3).

As in the previous analysis, the LTBI diagnosis with TST with ≥10 mm and ≥5 mm cut-off points corresponded to lower cost and more effective approaches, although the first strategy proves to be more cost-effective. The single use of the QFT-GIT has an incremental cost of over US$ 130 thousand dollars, with less effectiveness regarding the number of TB cases avoided with TST with ≥5 mm cut-off point (Table 3).

**Sensitivity analysis.** Tornado diagrams were developed to examine the the variables that most impacted the results of the decision models. Regardless of the assessed outcome measure, the parameters that showed a greater impact on the ratio of cost-effectiveness were the rate of adherence to LTBI treatment, the cost of LTBI treatment with INH for 6 months, and the prevalence of recent LTBI. (Figs 3 and 4). Costs of TST and costs of LTBI diagnosis also influenced the results. Other variables did not substantially affect the results.

Given the lack of information on adherence rate after LTBI diagnosis with QTF-GIT, bivariate sensitivity analysis was performed simulating different adhesion probabilities for the examined tests (ranging from 0.43 to 0.78), with incremental cost-effectiveness ratio remained consistently favorable to TST with 10mm cut-off point for both outcomes studied. Equally favorable results for the ≥10mm TST-based screening strategy were observed in bivariate analysis performed with QTF GIF at its lower cost limit and strategies using TST at the upper threshold.

Considering the outcome measure number of individuals correctly classified by the tests, the strategy corresponding to TST with 5mm cut-off point becomes the most cost-effective

**Table 3. Cost-effectiveness of screening strategies for diagnosis of Latent Tuberculosis Infection using the number of cases of active tuberculosis avoided as measure of effectiveness.**

| Diagnostic Strategies | Total Cost (US $) | Incremental Cost (US $) | Effectiveness | Incremental Effectiveness | Cost/ individuals correctly classified (US$) | ICER (US$) |
|---|---|---|---|---|---|---|
| Comparison of diagnostic strategies for LTBI in relation to the strategy TST≥10 mm[a] | | | | | | |
| TST ≥10 mm | 30,084.47 | _ | 3,881 | _ | 7.75 | _ |
| TST ≥5 mm | 30,471.02 | 386.55 | 3,886 | 5 | 7.84 | 77.31 |
| TST ≥5 mm/ QFT-GIT | 46,897.44 | 16,812.97 | 3,869 | -12 | 12.12 | dominated |
| TST ≥10 mm/ QFT-GIT | 47,376.73 | 17,292.26 | 3,865 | -16 | 12.25 | dominated |
| QFT-GIT | 160,531.88 | 130,447.41 | 3,884 | 3 | 41.33 | dominated |
| Comparison of diagnostic strategies for LTBI in relation to each other [b] | | | | | | |
| TST ≥10 mm | 30,084.47 | _ | 3,881 | _ | 7.75 | _ |
| TST ≥5 mm | 30,471.02 | 386.55 | 3,886 | 5 | 7.84 | 77.31 |
| TST ≥5 mm/ QFT-GIT | 46,897.44 | 16,426.42 | 3,869 | -17 | 12.12 | dominated |
| TST ≥10 mm/ QFT-GIT | 47,376.73 | 479.29 | 3,865 | -4 | 12.25 | dominated |
| QFT-GIT | 160,531.88 | 130,060.86 | 3,884 | -2 | 41.33 | dominated |

Caption: ICER–Incremental Cost-Effectiveness Ratio; LTBI–Latent Tuberculosis Infection; QFT-GIT–QuantiFERON-TB Gold In-Tube; TST–Tuberculosis Skin Test.

Notes:

[a]–The ICER was estimated based on TST strategy with the ≥10 mm cut-off point, only the dominated strategy was demarcated, without removing it from the table.

[b]–The ICER was estimated by comparing LTBI diagnostic strategies with each other. Incremental Costs was calculated by the difference between the cost of the strategy and the cost of its previous strategy and Incremental Effectiveness calculated by the difference between the effectiveness of the strategy and the effectiveness of its previous strategy. US$ 1.00 = R$ 2.35 (mean exchange rate in 2014).

when TST sensitivity for 10mm cut-off increases to the upper limit of the range. In this situation, even experiencing a growth in the number of individuals correctly classified (from 6,395 to 7,090), the associated costs increase substantially. Changes are also observed in the most cost-effective strategy when the sensitivity of the TST with ≥5 mm cut-off point is reduced to the lower limit of the variation parameter range. Although the costs are reduced by US$ 2,413.67, there is a difference of 541 individuals correctly classified (Table 4). For all other variables, the strategy currently recommended by the NTBP of using ≥10mm TST is the most cost-effective.

Regarding the number of new TB cases avoided, it was observed that the main variables that altered the cost-effectiveness ratio were reduced the sensitivity of ≥5 mm TST (0.82 to 0.68) and reducing the specificity of ≥10mm TST (0.59 to 0.43). In both cases, the change in the most cost-effective strategy was the result of a substantial increase in costs, especially those resulting from therapeutic intervention and its consequences in terms of severe DILI, hospitalization, and death. For any variation in the probability of returning to TST reading, the results were in agreement with the reference case, i.e. TST with 10 mm cut-off was the most cost-effective strategy. The same applies to variations in all other parameters of the model.

## Discussion

A cost-effectiveness model, using TST and QFT-GIT tests, was developed to assess the costs and effectiveness of five strategies for diagnosis and treatment of LTBI among primary HCWs at risk of tuberculosis. Our results of this research showed the strategy based on TST with 10

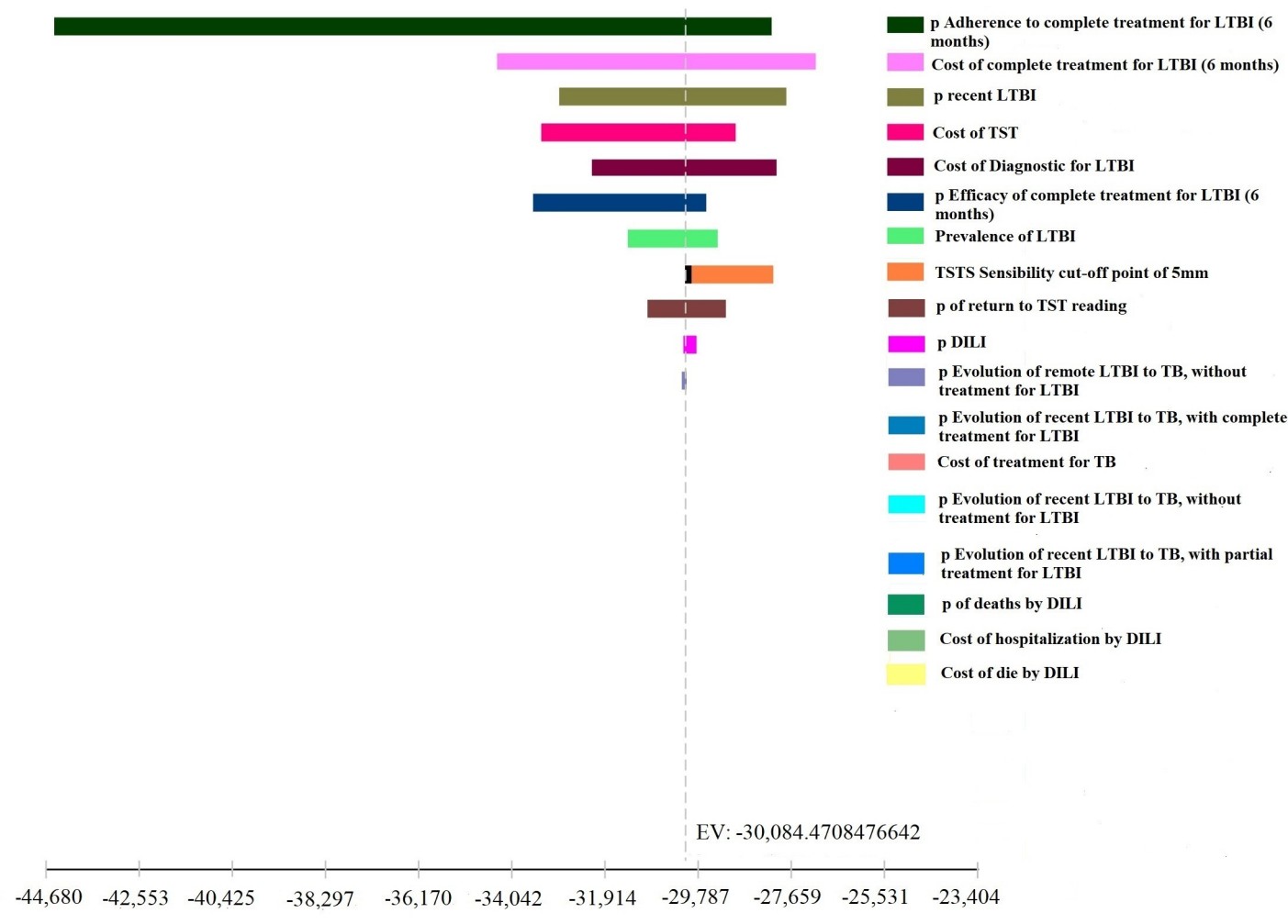

**Fig 3. Sensitivity analysis in Tornado chart: Number of individuals correctly classified by the tests.**

mm cut-off point as the most cost-effective at a cost of US$ 4.70 per case correctly classified and US$ 7.75 per case of active tuberculosis avoided.

The most favorable result to the TST was consistent with the findings of two studies. The study conducted by Steffen et al.[26] was carried out considering the Brazilian public system health, however, the authors used TB contacts as their study sample, which is why the researchers had lower prevalence of LTBI (0.35, range from 0.20 to 0.65). Three strategies for LTBI diagnosis were examined in this cost-effectiveness analysis, using a decision-analytic model: TST with cut-off of ≥5 mm, QuantiFERON®-TB Gold In-Tube (QFT-GIT) and TST-positive results confirmed by QFT-GIT (TST + / QFT-GIT). The outcome measure used was the number of cases of tuberculosis avoided in two years, and the accuracy parameters of the tests were similar to the present study. The costs of tests examined in the Steffen's study, however, were higher than the values used here, US$ 10.56 for the TST (versus US$ 7.62) and US$ 48.26 for the QFT-GIT (versus the US$ 39.00). TST proved to be the most cost-effective strategy (US$ 16.021/case averted). Mancuso et al.[60] examined the cost-effectiveness of nine different screening strategies for LTBI in U.S. military recruits, including TST, T-SPOT®.TB,

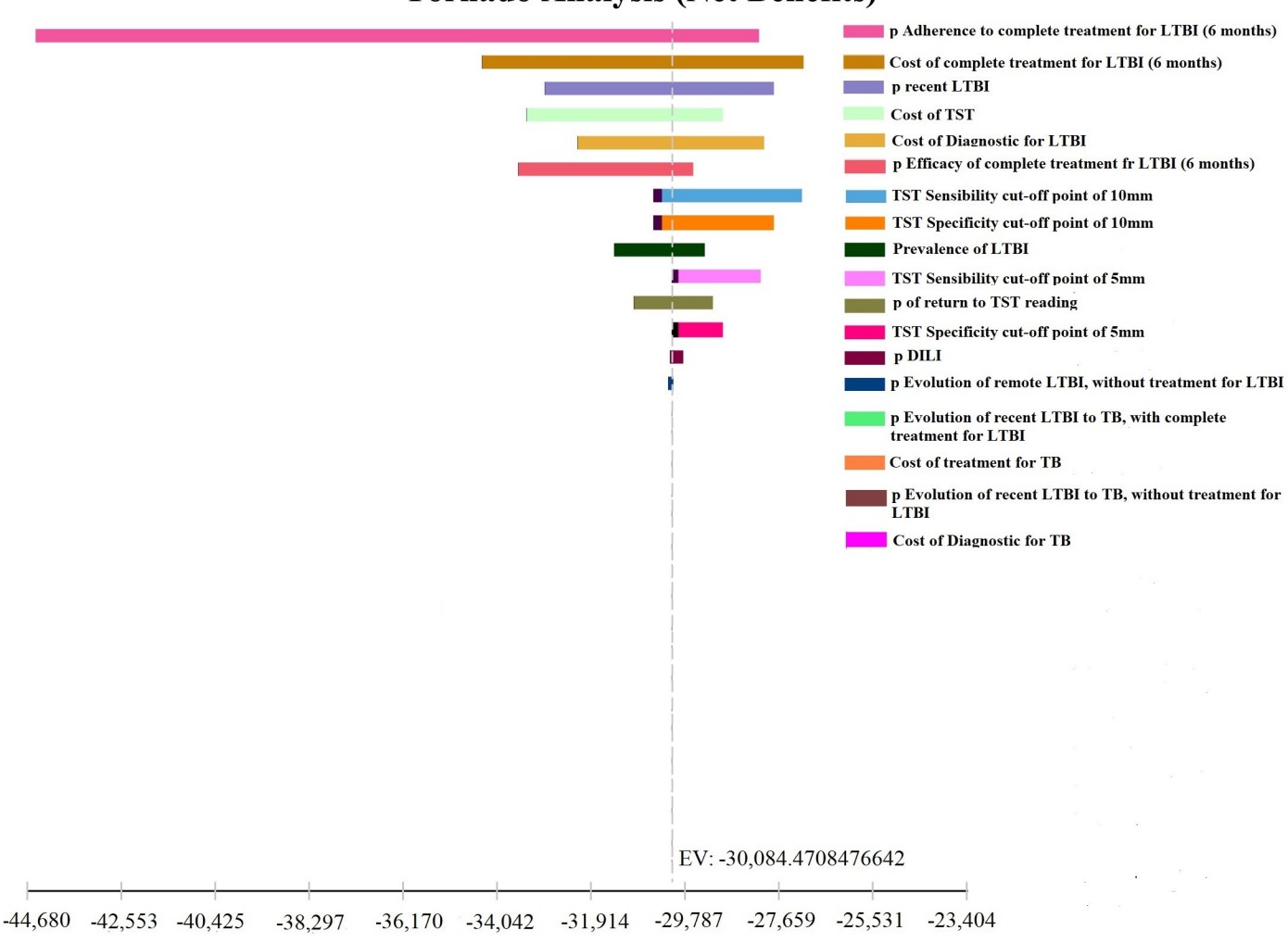

**Fig 4. Sensitivity analysis in Tornado chart: Number of cases of active tuberculosis avoided.**

QuantiFERON®-TB Gold In-Tube alone and combinations of these tests with universal testing or based on of risk assessment questionnaires. The cut-off point used for the TST is not clear. The sensitivity and specificity parameters used were, respectively, 0.77 (0.70–0.99) and 0.99 (0.95–1.00). The health outcome measured in this analysis was cases of active TB prevented. The societal perspective was taken over a 20-year analytic horizon, with discount in future costs at 3% annually. Targeted testing using TST was slightly more cost-effective than targeted testing using either QTF-GIT or T-SPOT, but these estimates were very sensitive to changes in model assumptions.

Results of our study differ from other previous cost-effectiveness analyzes, which were favorable to QTF-GIT. Diel et al.[55] already indicated that the combination of the QFT-G assay following the TST screening of close-contacts at a cutoff induration size of 5 mm was more cost-effective in Germany, followed by the QFT-GIT strategy alone. Kowada and Marra showed that in populations vaccinated with BCG, single use of QFT-GIT assay was the most cost-effective strategy for TB contacts in Japan and Canada, respectively[61, 62]. The study by de Perio et al.[63] also examined the cost-effectiveness of the new IGRA in detecting LTBI in

**Table 4. Univariate sensitivity analysis of the comparison of diagnostic strategies for Latent Tuberculosis Infection in relation to TST≥10 mm strategy, considering as outcome measure the number of individuals correctly classified and the number of active tuberculosis cases averted by the diagnostic tests for Latent Tuberculosis Infection.**

| Diagnostic Strategies | Total Cost (US$) | Incremental Cost (US$) | Effectiveness | Incremental Effectiveness | ICER (US$) |
|---|---|---|---|---|---|
| *Number of individuals correctly classified by the tests* | | | | | |
| **Base Case** | | | | | |
| TST≥10 mm | 30,084.47 | _ | 6,395 | _ | _ |
| TST≥5 mm | 30,471.02 | 386.55 | 6,704 | 309 | 1.25 |
| TST≥5 mm/QFT-GIT | 46,897.44 | 16,426.42 | 8,222 | 1,518 | 10.82 |
| TST≥10 mm/QFT-GIT | 47,376.73 | 479.29 | 8,069 | -153 | dominated |
| QFT-GIT | 160,531.88 | 113,155.15 | 9 | 931 | 121.54 |
| **TST sensitivity with ≥10 mm cut-off point = 55%** | | | | | |
| TST ≥10 mm | 27,132.40 | _ | 5,545 | _ | _ |
| TST ≥5 mm | 30,471.01 | 3,338.61 | 6,704 | 1,159 | 2.88 |
| TST ≥10 mm/QFT-GIT | 42,753.92 | 12,282.91 | 7,406 | 702 | 17.49 |
| TST ≥5 mm/QFT-GIT | 46,897.44 | 4,143.52 | 8,222 | 816 | 5.07 |
| QFT-GIT | 160,531.88 | 113,634.44 | 9 | 778 | 146.05 |
| **TST sensitivity with ≥10 mm cut-off point = 95%** | | | | | |
| TST ≥5 mm | 30,471.01 | _ | 6,704 | _ | _ |
| TST≥10 mm | 34,028.57 | 3,557.56 | 7,09 | 386 | 9.21 |
| TST ≥5 mm/QFT-GIT | 46,897.44 | 12,868.87 | 8,222 | 1,132 | 11.36 |
| TST ≥10 mm/QFT-GIT | 53,097.51 | 6,200.07 | 8,612 | 390 | 15.89 |
| QFT-GIT | 160,531.88 | 107,434.37 | 9 | 388 | 276.89 |
| **TST sensitivity with ≥5 mm cut-off point = 68%** | | | | | |
| TST ≥5 mm | 28,057.35 | _ | 6,163 | _ | _ |
| TST ≥10 mm | 30,084.47 | 2,027.12 | 6,395 | 232 | 8.73 |
| TST ≥5 mm/QFT-GIT | 43,278.68 | 13,194.21 | 7,8 | 1,405 | 9.39 |
| TST ≥10 mm/QFT-GIT | 47,376.73 | 4,098.05 | 8,069 | 269 | 15.23 |
| QFT-GIT | 160,531.88 | 113,155.15 | 9 | 931 | 121.54 |
| **TST sensitivity with ≥5 mm cut-off point = 97%** | | | | | |
| TST ≥10 mm | 30,084.47 | _ | 6,395 | _ | _ |
| TST ≥5 mm | 33,980.72 | 3,896.25 | 7,284 | 889 | 4.38 |
| TST ≥10 mm/QFT-GIT | 47,376.73 | 13,396.01 | 8,069 | 785 | 17.06 |
| TST ≥5 mm/QFT-GIT | 51,947.96 | 4,571.23 | 8,674 | 605 | 7.55 |
| QFT-GIT | 160,531.88 | 108,583.92 | 9 | 326 | 333,07 |
| *Number of cases of active tuberculosis avoided* | | | | | |
| **Base Case** | | | | | |
| TST ≥10 mm | 30,084.47 | _ | 3,881 | _ | _ |
| TST ≥5 mm | 30,471.02 | 386.55 | 3,886 | 5 | 77.31 |
| TST ≥5 mm/QFT-GIT | 46,897.44 | 16,426.42 | 3,869 | -17 | dominated |
| TST ≥10 mm/QFT-GIT | 47,376.73 | 479.29 | 3,865 | -4 | dominated |
| QFT-GIT | 160,531.88 | 113,155.15 | 3,884 | 19 | 5,955.53 |
| **TST sensitivity with ≥10 mm cut-off point = 55%** | | | | | |
| TST ≥10 mm | 27,132.40 | _ | 3,86 | _ | _ |
| TST ≥5 mm | 30,471.01 | 3,338.61 | 3,886 | 26 | 128.40 |
| TST ≥10 mm/QFT-GIT | 42,758.17 | 12,287.16 | 3,849 | -37 | dominated |
| TST ≥5 mm/QFT-GIT | 46,897.44 | 4,139.27 | 3,869 | 20 | 206.96 |
| QFT-GIT | 160,531.88 | 113,634.44 | 3,884 | 15 | 7,575.62 |
| **TST sensitivity with ≥10 mm cut-off point = 95%** | | | | | |
| TST ≥5 mm | 30,471.01 | _ | 3,886 | _ | _ |

*(Continued)*

**Table 4.** (Continued)

| Diagnostic Strategies | Total Cost (US$) | Incremental Cost (US$) | Effectiveness | Incremental Effectiveness | ICER (US$) |
|---|---|---|---|---|---|
| TST ≥10 mm | 34,028.57 | 3,557.56 | 3,898 | 12 | 296.46 |
| TST ≥5 mm/QFT-GIT | 46,897.44 | 12,868.87 | 3,869 | -29 | dominated |
| TST ≥10 mm/QFT-GIT | 53,097.51 | 6,200.07 | 3,878 | 9 | 688.89 |
| QFT-GIT | 160,531.88 | 107,434.37 | 3,884 | 6 | 17,905.72 |
| **TST sensitivity with ≥5 mm cut-off point = 68%** | | | | | |
| TST ≥5 mm | 28,057.35 | _ | 3,872 | _ | _ |
| TST ≥10 mm | 30,084.47 | 2,027.12 | 3,881 | 9 | 225.23 |
| TST ≥5 mm/QFT-GIT | 43,278.67 | 13,194.20 | 3,858 | -23 | dominated |
| TST ≥10 mm/QFT-GIT | 47,376.73 | 4,098.06 | 3,865 | 7 | 585.43 |
| QFT-GIT | 160,531.88 | 113,155.15 | 3,884 | 19 | 5,955.53 |
| **TST sensitivity with ≥5 mm cut-off point = 97%** | | | | | |
| TST ≥10 mm | 30,084.47 | _ | 3,881 | _ | _ |
| TST ≥5 mm | 33.980.72 | 3,896.25 | 3,9 | 19 | 205.06 |
| TST ≥10 mm/QFT-GIT | 47,376.73 | 13,396.01 | 3,865 | -35 | dominated |
| TST ≥5mm/QFT-GIT | 51,947.96 | 4,571.23 | 3,88 | 15 | 304.74 |
| QFT-GIT | 160,531.88 | 108,583.92 | 3,884 | 4 | 27,145.98 |
| **TST specificity with ≥10 mm cut-off point = 43%** | | | | | |
| TST ≥5 mm | 30,471.01 | _ | 3,886 | _ | _ |
| TST ≥10 mm | 37,527.68 | 7,056.67 | 3,881 | -5 | dominated |
| TST ≥5 mm/QFT-GIT | 46,897.44 | 9,369.76 | 3,869 | -12 | dominated |
| TST ≥10 mm/QFT-GIT | 67,776.09 | 20,878.65 | 3,865 | -4 | dominated |
| QFT-GIT | 160,531.88 | 92,755.79 | 3,884 | 19 | 4,881.88 |
| **TST specificity with ≥10 mm cut-off point = 73%** | | | | | |
| TST ≥10 mm | 27,764.41 | _ | 3,881 | _ | _ |
| TST ≥5 mm | 30,471.01 | 2,706.60 | 3,886 | 5 | 541.32 |
| TST ≥10 mm/QFT-GIT | 37,190.45 | 6,719.44 | 3,865 | -21 | dominated |
| TST ≥5 mm/QFT-GIT | 46,897.44 | 9,706.99 | 3,869 | 4 | 2,426.74 |
| QFT-GIT | 160,531.88 | 113,634.44 | 3,884 | 15 | 7,575.62 |
| **TST specificity with ≥5 mm cut-off point = 35%** | | | | | |
| TST≥10 mm | 30,084.47 | - | 3,881 | 0 | - |
| TST≥5 mm | 44,123.57 | 14,039.09 | 3,886 | 5 | 2,807.82 |
| TST≥10 mm/QFT-GIT | 47,376.73 | 3,253.17 | 3,865 | -21 | dominated |
| TST≥5 mm/QFT-GIT | 82,892.72 | 38,769.16 | 3,869 | -17 | dominated |
| QFT-GIT | 160,531.90 | 116,408.30 | 3,884 | -2 | dominated |
| **TST specificity with ≥5 mm cut-off point = 79%** | | | | | |
| TST≥5 mm | 28,925.61 | - | 3,886 | 0 | _ |
| TST≥10 mm | 30,084.47 | 1,158.86 | 3,881 | -5 | dominated |
| TST≥5 mm/QFT-GIT | 36,428.17 | 7,502.55 | 3,869 | -17 | dominated |
| TST≥10 mm/QFT-GIT | 47,376.73 | 18,451.12 | 3,865 | -21 | dominated |
| QFT-GIT | 160,531.90 | 131,606.30 | 3,884 | -2 | dominated |

Caption: ICER–Incremental Cost-Effectiveness Ratio; LTBI–Latent Tuberculosis Infection; QFT-GIT–QuantiFERON-TB Gold In-Tube; TST–Tuberculosis Skin Test.

health professionals, comparing QFT-G, QFT-GIT, and TST in TB low-incidence countries. Their results pointed out that both IGRAs were more effective and less costly than the TST, whether or not the HCW had been previously vaccinated with BCG. Their findings, however, are not suitable for comparison to the present paper due to, among other things, the type of

model and perspective (Markov state-transition decision analytic model using the societal perspective) and the measure of effectiveness used (QALYs). Other significant differences include the fact that it was conducted in a country with low incidence of tuberculosis and focused on hospital health workers.

The inclusion of IGRAs has advanced the diagnosis of tuberculosis significantly and has been recommended in recent years as a potential replacement for the TST[4, 24]. Centers for Disease Control and Prevention (CDC) recommend the use of IGRA in all circumstances in which TST is currently used, including in health professionals[64]. In contrast, a guideline published by the National Institute for Health and Care Excellence (NICE) in 2006, in the United Kingdom, recommends its use as a sequential test, restrictively to individuals at risk of LTBI (children, people who are immunocompromised or at risk of immunosuppression and people from countries with a high incidence of TB) and in those with positive results for TST [28]. The Canadian guidelines on IGRAs have not recommended their use for serial testing of HCWs[65, 66], indicating that its use of IGRAs for routine screening of HCWs remains a controversy matter.

Regarding the increasing use of the QFT in developed countries and their use in the private sector of Brazilian health, it is still bought by high costs in the international market.

It is well known that IGRAs have higher sensitivity and specificity than TST[35, 38], leads to lower false-positive results and avoids unnecessary treatment of LTBI[67], and do not produce booster phenomenon. This assay require only a single visit to the health service for its realization and may increase adherence to isoniazid treatment by health professionals[68].

However, the QFT-GIT trial is more expensive than the TST test (US$ 39.00 vs. US$ 7.62). Even when the model is simulated with the upper limit of the cost of TST range (US$ 9.54), and the cost of IGRA at its lowest estimated value (US$ 31.77), this strategy continues to be the most cost-effective for the diagnosis of LTBI among health professionals for both health outcomes studied, because the cost differences between the two tests are very wide. The significant difference in the cost of testing and the high prevalence of latent tuberculosis in Brazil are factors that may explain the absence of favorable cost-effectiveness to the QTF, regardless of whether used alone or sequentially. In addition, the variation of the QTF-GIT accuracy measures did not make the strategies related to this technology more cost-effective under any circumstances. In turn, changes in TST accuracy parameters produce a change in the cut-off points for the TST that are more cost-effective.

For this analysis, the use of isoniazid prophylaxis was defined for six months because it is the recommended regimen according to current guidelines in Brazil[30] and is considered preferable in relation to cost-effectiveness[69]. Although efficacy results of LTBI treatment according to the duration of the preventive regimen are scarce in the literature, a Union trial indicated that increasing the treatment time from six to 12 months does not substantially increases efficacy but reduces therapy adherence[32].

It is important to note that the overall cost of LTBI treatment has the costs of DILI incorporated. The literature shows that hepatotoxicity associated with the treatment of LTBI with INH is not an ordinary event, especially regarding severe disease[46–48]. Since the probability of severe adverse reactions in the treated population is small, the impact of the DILI on overall costs was minor. Moreover, the overall costs of treating LTBI were reduced due to the shorter period of treatment.

It is well known that IGRAs have higher sensitivity and specificity than TST[35, 38], do not produce booster phenomenon and require only a single visit to the health service for its realization. However, the variation of the IGRAs accuracy measures did not make the strategies related to this technology more cost-effective under any circumstances. In turn, changes in

TST accuracy parameters produce a change in the cut-off points for the TST that are more cost-effective.

The prevalence of LTBI was a parameter that often proves to be relevant in previous studies, usually estimated based on latent infection surveys using TST in HCWs with wide variation in results[70–80]. In the present study, the prevalence of LTBI was estimated by INATA survey (a study in primary health units in all regions of Brazil), and its range (36 to 44%) was used in the sensitivity analysis, without impacting the results of the cost-effeteness analysis.

Some limitations of this study should be discussed. Even using results of a national survey, to consider a single source for the parameter of LTBI prevalence in HCW can be regarded as a study limitation. Costs for repeated TST in the case of lost reading were not considered, but the rate of no return was estimated as less than 5%.

The assumption is that all cases of active TB and LTBI were sensitive to antituberculosis drugs used because multidrug-resistant tuberculosis prevalence is still low (lower than 1,5%) in Brazil. In this evaluation, we decided to use intermediate outcomes and a short time horizon, reinforcing the importance of further studies that incorporate more finalistic outcomes such as survival and quality of life. TST has been used for over a century and has shown benefits in the treatment of LTBI in situations in which the TST positive is well defined[81, 82]. However, it is worth noting that the training for TST is time-consuming, complex and can generate additional costs that were not considered in this study. Despite the clarity of a simplified test algorithm, the convenience of fewer return visits and the clinical benefits of fewer false-positive results (avoids costs of treatment of LTBI unnecessary), IGRAs still have gaps to detect recent infection of TB[83], and show a large difference in costs between TST≥10 mm and QFT-GIT strategies.

The current analysis suggests that the TST constitutes the LTBI screening strategy as cost-effective in the Brazilian scene, even after a significant reduction in QFT-GIT costs and despite the high number of patients undergoing treatment for LTBI. Further studies on cost-effectiveness of this new technology are needed.

## Supporting information

**S1 Checklist. CHEERS.** Researcher Checklist.
(PDF)

## Author Contributions

**Conceptualization:** Rafaela Borge Loureiro, Ethel Leonor Noia Maciel, Rosangela Caetano, José Ueleres Braga.

**Data curation:** Rafaela Borge Loureiro, Renata Lyrio Peres, Geisa Fregona.

**Formal analysis:** Rafaela Borge Loureiro, Rosangela Caetano, José Ueleres Braga.

**Funding acquisition:** Ethel Leonor Noia Maciel.

**Investigation:** Rafaela Borge Loureiro, Ethel Leonor Noia Maciel, Rosangela Caetano, José Ueleres Braga.

**Methodology:** Rafaela Borge Loureiro, Ethel Leonor Noia Maciel, Rosangela Caetano, José Ueleres Braga.

**Project administration:** Rafaela Borge Loureiro, Ethel Leonor Noia Maciel.

**Resources:** Ethel Leonor Noia Maciel.

**Software:** Rafaela Borge Loureiro, Ethel Leonor Noia Maciel, Rosangela Caetano.

**Supervision:** Rafaela Borge Loureiro, Ethel Leonor Noia Maciel, Rosangela Caetano, José Ueleres Braga.

**Validation:** Rafaela Borge Loureiro, Ethel Leonor Noia Maciel, Rosangela Caetano, José Ueleres Braga.

**Visualization:** Rafaela Borge Loureiro, Ethel Leonor Noia Maciel, Rosangela Caetano, José Ueleres Braga.

**Writing – original draft:** Rafaela Borge Loureiro, Ethel Leonor Noia Maciel, Rosangela Caetano, Jonathan E. Golub, José Ueleres Braga.

**Writing – review & editing:** Rafaela Borge Loureiro, Ethel Leonor Noia Maciel, Rosangela Caetano, Jonathan E. Golub, José Ueleres Braga.

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
