## [Decision Letter · Decision Letter 0]

17 Jul 2019

PONE-D-19-15042

Cost-Effectiveness of QuantiFERON®-TB Gold In-Tube Versus Tuberculin Skin Test for Diagnosis and Treatment of Latent Tuberculosis Infection in Primary Health Care Workers in Brazil

PLOS ONE

Dear Dr. Braga,

Thank you for submitting your manuscript to PLOS ONE. After careful consideration, we feel that it has merit but does not fully meet PLOS ONE’s publication criteria as it currently stands. Therefore, we invite you to submit a revised version of the manuscript that addresses the points raised during the review process.

We would appreciate receiving your revised manuscript by Aug 31 2019 11:59PM. To enhance the reproducibility of your results, we recommend that if applicable you deposit your laboratory protocols in protocols.io, where a protocol can be assigned its own identifier (DOI) such that it can be cited independently in the future. For instructions see: http://journals.plos.org/plosone/s/submission-guidelines#loc-laboratory-protocols

We look forward to receiving your revised manuscript.

Kind regards,

Frederick Quinn

Academic Editor

PLOS ONE

Journal Requirements:

Reviewers' comments:

Reviewer's Responses to Questions

**Comments to the Author**

1. Is the manuscript technically sound, and do the data support the conclusions?

Reviewer #1: Yes

Reviewer #2: Yes

2. Has the statistical analysis been performed appropriately and rigorously? 

Reviewer #1: Yes

Reviewer #2: Yes

3. Have the authors made all data underlying the findings in their manuscript fully available?

Reviewer #1: Yes

Reviewer #2: Yes

4. Is the manuscript presented in an intelligible fashion and written in standard English?

Reviewer #1: No

Reviewer #2: No

5. Review Comments to the Author

Reviewer #1: This was an interesting and informative study comparing cost effectiveness of 5 strategies for diagnosis of latent TB in health care workers in Brazil. The evaluation was based on a hypothetical cohort of 10,000 HCW. Outcomes evaluated included number of individuals correctly classified and number of TB cases avoided. The conclusions of the study were that TST with a ≥10 mm cutoff was the most cost saving strategy for diagnosis of latent TB in this population.

Overall: The text is a bit difficult to read with some issues with word order, etc that I believe are a byproduct of the translation to English. I would suggest having a native English speaker edit the manuscript if possible.

Minor points:

Authors use both 10 mm and ≥10 mm throughout the manuscript. ≥10 mm should be used consistently.

Line 49: Suggest editing to say “preventing active disease” or “identification of latent and active disease in an effort to reduce transmission”

Lines 51-53: Suggest updating to more recent statistics.

Lines 118-122: This text is repetitive with the figure legends and unnecessary.

Line 129: Please edit title of Figure 2 so that it is not identical to the title of Figure 1.

373-375: The effectiveness of all strategies in regard to number of tuberculosis cases avoided is very similar. Is the difference between the strategies statistically significant? Mention should be made of the closeness of all strategies for this measure of outcome effectiveness.

Line 440: “…the change in the most cost-effective strategy was a result of the substantial increase in costs.” Suggest describing which costs were increased.

Line 494: define STF

Table 1: I do not understand the calculation for “Cost of Complete LTBI Treatment(6months)”, “Cost of Partial LTBI Treatment (3 months)” or “Cost of Active TB Treatment”. Why do line items for the drug(s) used for treatment have “(month)” beside them? Does this mean that this is the cost for one month? If so, the calculations for the total cost of treatment are incorrect since the values have only been included once. If these values actually represent the cost of drug for the entire duration of treatment why do they have “(month)” beside them?

Discussion: There is no discussion of the comparable results for all the tests for number of cases avoided.

Other Suggestions:

• I think that the addition of number-needed-to treat (the number of people treated for LTBI to prevent an active TB case) for each strategy would be an interesting and valuable addition to the study.

• Authors mention that adherence to LTBI treatment by HCW is low due to the known false-positivity rate of TST. Is the adherence rate to LTBI treatment for individuals diagnosed with QFT known or can it be realistically estimated? Would it be possible to do a cost calculation for TST vs QFT using different adherence rates for each test? This could be really important since the rate of adherence to LTBI treatment was one of the most impactful variables in the cost effectiveness calculations and would also likely impact number of active TB cases avoided.

Reviewer #2: The paper present interesting data investigating the cost effectiveness of QuantiFERON®-TB Gold In-Tube and Tuberculin Skin Test for Diagnosis and Treatment of Latent Tuberculosis Infection in Primary Health Care Workers in Brazil. The method section is well explained, however the result and discussion sections need to be restructured in a better way.

The tables in the results section (especially table 4) are too large, they need to be presented in a more efficient way. a summarizing table can be used and the rest of the data can be attached to the paper as supporting information.

The manuscript need to be English edited in a very detailed way. Many of the sentences are too long and contain grammar and formulation mistakes.

Some suggestions and comments below:

Line 59: add more recent statistics

Line 65: is considered as a priority by WHO in controlling the disease ??

Line 138: do you mean true positives and true negatives?

Line 181: cut-off

184-185: data based on...

351-352: column 2 in table 2, is it the total cost, if so mention it

453: what do you mean by study contacts?

471: are you comparing the study results to other studies performed outside of Brazil? if so, find a good way to make the transition between this paragraph and the previous one

549-550: this paragraph is out of context, you will need to develop it further and find a way to link it to the previous paragraph (or you can delete it)

6. PLOS authors have the option to publish the peer review history of their article (what does this mean?). If published, this will include your full peer review and any attached files.

Reviewer #1: No

Reviewer #2: No

---

## [Author Response · Author response to Decision Letter 0]

15 Oct 2019

PONE-D-19-15042

Cost-Effectiveness of QuantiFERON®-TB Gold In-Tube Versus Tuberculin Skin Test for Diagnosis and Treatment of Latent Tuberculosis Infection in Primary Health Care Workers in Brazil

Dear Editors,

The authors are grateful for the suggestions of the reviewers, who point out as very important for the manuscript improvement.

Please also be advised that changes and additions to version 2 text are marked as requested and an unmarked version of revised paper without tracked changes has been included (labeled 'Manuscript').

A summary of the changes made in response to the requests is provided below to facilitate the evaluation of the achievement.

Regards,

Jose Ueleres Braga 

Review Comments to the Author

Reviewer #1: 

This was an interesting and informative study comparing cost effectiveness of 5 strategies for diagnosis of latent TB in health care workers in Brazil. The evaluation was based on a hypothetical cohort of 10,000 HCW. Outcomes evaluated included number of individuals correctly classified and number of TB cases avoided. The conclusions of the study were that TST with a ≥10 mm cutoff was the most cost saving strategy for diagnosis of latent TB in this population.

Overall: 

The text is a bit difficult to read with some issues with word order, etc that I believe are a byproduct of the translation to English. I would suggest having a native English speaker edit the manuscript if possible.

R.: Yes. We reviewed the entire manuscript with a native English speaker, looking to correct the issues identified.

Minor points:

Authors use both 10 mm and ≥10 mm throughout the manuscript. ≥10 mm should be used consistently.

R.: We agree. We replace the expression 10mm with ≥10mm when used to classify individuals by Mtb infection status.

Line 49: Suggest editing to say “preventing active disease” or “identification of latent and active disease in an effort to reduce transmission”

R.: The text of the manuscript was modified to comply with the suggestion made by the reviewer.

Lines 51-53: Suggest updating to more recent statistics.

R.: The text has been modified to include the latest tuberculosis statistics in Brazil and worldwide, with appropriate replacement in the reference section.

Lines 118-122: This text is repetitive with the figure legends and unnecessary.

R.: We agree with the reviewer's comment and the text has been rewritten to eliminate the pointed repetition.

Line 129: Please edit title of Figure 2 so that it is not identical to the title of Figure 1.

R.: We agree. The titles of figures 1 and 2 have been edited to clarify what is in place. Unnecessary repetitions when displayed in text were deleted.

373-375: The effectiveness of all strategies in regard to number of tuberculosis cases avoided is very similar. Is the difference between the strategies statistically significant? Mention should be made of the closeness of all strategies for this measure of outcome effectiveness.

R.: We agree. The manuscript was revised to highlight the closeness of the effectiveness values. No statistical tests were applied to compare these figures

Line 440: “…the change in the most cost-effective strategy was a result of the substantial increase in costs.” Suggest describing which costs were increased. 

R.: The text was rewritten to include the suggestion of the reviewer.

Line 494: define STF 

R.: We apologize for the typo. The correct acronym is QTF

Table 1: I do not understand the calculation for “Cost of Complete LTBI Treatment(6months)”, “Cost of Partial LTBI Treatment (3 months)” or “Cost of Active TB Treatment”. Why do line items for the drug(s) used for treatment have “(month)” beside them? Does this mean that this is the cost for one month? If so, the calculations for the total cost of treatment are incorrect since the values have only been included once. If these values actually represent the cost of drug for the entire duration of treatment why do they have “(month)” beside them?

R.: We appreciate the remark. The values in table 1 correspond to the total cost of treatment. The term “month” has been removed to avoid misunderstanding.

Discussion: 

There is no discussion of the comparable results for all the tests for number of cases avoided.

R.: The discussion section has been rewritten and this aspect has been covered.

Other Suggestions:

• I think that the addition of number-needed-to treat (the number of people treated for LTBI to prevent an active TB case) for each strategy would be an interesting and valuable addition to the study.

R.: We welcome the suggestion, but understand that the epidemiological measures used are considered appropriate for this cost-effectiveness study.

• Authors mention that adherence to LTBI treatment by HCW is low due to the known false-positivity rate of TST. Is the adherence rate to LTBI treatment for individuals diagnosed with QFT known or can it be realistically estimated? Would it be possible to do a cost calculation for TST vs QFT using different adherence rates for each test? This could be really important since the rate of adherence to LTBI treatment was one of the most impactful variables in the cost effectiveness calculations and would also likely impact number of active TB cases avoided.

R.: There were no published studies on adherence to LTBI treatment after QTF screening at the time of development of the current study. A literature search to answer the reviewer found few updated studies addressing treatment adherence rate after QTF testing, and their values are compatible with the parameters assumed in this study. A bivariate sensitivity analysis was performed considering different adherence rates after diagnosis when each test was used, and its result did not indicate a change in the conclusion, in favor of the TST strategy with 10mm cut-off. There were insignificant changes in the number of active TB cases avoided by applying different LTBI treatment adherence rates according to the diagnostic strategy employed.

Reviewer #2: 

The paper present interesting data investigating the cost effectiveness of QuantiFERON®-TB Gold In-Tube and Tuberculin Skin Test for Diagnosis and Treatment of Latent Tuberculosis Infection in Primary Health Care Workers in Brazil. The method section is well explained, however the result and discussion sections need to be restructured in a better way.

The tables in the results section (especially table 4) are too large, they need to be presented in a more efficient way. a summarizing table can be used and the rest of the data can be attached to the paper as supporting information.

The manuscript need to be English edited in a very detailed way. Many of the sentences are too long and contain grammar and formulation mistakes.

R.: Thanks for the suggestion, we've reformatted tables to make your look cleaner. However, we believe it would be relevant for the reader to keep part of the information present in the manuscript in its original format. Following the suggestions we also rewrote the result and discussion sections to make the content clearer. Finally, the manuscript was reviewed by a native American to minimize errors with the use of the English language. 

Some suggestions and comments below:

Line 59: add more recent statistics

R.: The text has been modified to include the latest tuberculosis statistics in Brazil and worldwide, with appropriate replacement in the reference section.

Line 65: is considered as a priority by WHO in controlling the disease ??

R.: Yes. WHO considers the identification of LTBI and its treatment in priority groups as an important strategy for disease control, and this view holds in the latest report available (2018).

Line 138: do you mean true positives and true negatives?

R.: Yes. The text has been modified to make it clear what was being said.

Line 181: cut-off

R.: We accept the suggestion and change the expression cutoff for cut-off throughout the text.

184-185: data based on...

R.: We apologize. The text has been corrected.

351-352: column 2 in table 2, is it the total cost, if so mention it

R.: We apologize. The text has been corrected.

453: what do you mean by study contacts?

R.: We revised the text to make it clearer..

471: are you comparing the study results to other studies performed outside of Brazil? if so, find a good way to make the transition between this paragraph and the previous one

R.: We thank the suggestion and rewrite the text, leaving separated the studies with concordant and discordant results from the evidenced in the present research.

549-550: this paragraph is out of context, you will need to develop it further and find a way to link it to the previous paragraph (or you can delete it)

R.: Thanks. We accept your suggestion.

---

## [Decision Letter · Decision Letter 1]

31 Oct 2019

Cost-Effectiveness of QuantiFERON®-TB Gold In-Tube Versus Tuberculin Skin Test for Diagnosis and Treatment of Latent Tuberculosis Infection in Primary Health Care Workers in Brazil

PONE-D-19-15042R1

Dear Dr. Braga,

We are pleased to inform you that your manuscript has been judged scientifically suitable for publication and will be formally accepted for publication once it complies with all outstanding technical requirements.

With kind regards,

Frederick Quinn

Academic Editor

PLOS ONE

Additional Editor Comments (optional):

Reviewers' comments:

Reviewer's Responses to Questions

**Comments to the Author**

1. If the authors have adequately addressed your comments raised in a previous round of review and you feel that this manuscript is now acceptable for publication, you may indicate that here to bypass the “Comments to the Author” section, enter your conflict of interest statement in the “Confidential to Editor” section, and submit your "Accept" recommendation.

Reviewer #1: All comments have been addressed

Reviewer #2: All comments have been addressed

2. Is the manuscript technically sound, and do the data support the conclusions?

Reviewer #1: (No Response)

Reviewer #2: Yes

3. Has the statistical analysis been performed appropriately and rigorously? 

Reviewer #1: (No Response)

Reviewer #2: Yes

4. Have the authors made all data underlying the findings in their manuscript fully available?

Reviewer #1: (No Response)

Reviewer #2: Yes

5. Is the manuscript presented in an intelligible fashion and written in standard English?

Reviewer #1: (No Response)

Reviewer #2: Yes

6. Review Comments to the Author

Reviewer #1: I suggest going over the manuscript again for correct English translation. Please check reference numbers throughout the manuscript for accuracy...I do not believe that the references indicated in lines 54 and 56 are the correct references for these statistics. Additionally, an analysis of the additional cost incurred by the high number of false-positives that would be inappropriately treated using the TST>=10mm strategy vs. other strategies and the total impact of this on the cost effectiveness would be beneficial.

Reviewer #2: I suggest to go through the manuscript one more time and correct the typos (e.g: 145-146: treatment of TB....)

7. PLOS authors have the option to publish the peer review history of their article (what does this mean?). If published, this will include your full peer review and any attached files.

Reviewer #1: No

Reviewer #2: No

---

## [Editor Report · Acceptance letter]

7 Nov 2019

PONE-D-19-15042R1 

Cost-Effectiveness of QuantiFERON®-TB Gold In-Tube Versus Tuberculin Skin Test for Diagnosis and Treatment of Latent Tuberculosis Infection in Primary Health Care Workers in Brazil 

Dear Dr. Braga:

I am pleased to inform you that your manuscript has been deemed suitable for publication in PLOS ONE. Congratulations! Your manuscript is now with our production department. 

With kind regards,

on behalf of

Dr. Frederick Quinn 

Academic Editor

PLOS ONE